# Seasonal Changes in Continuous Sedentary Behavior in Community—Dwelling Japanese Adults: A Pilot Study

**DOI:** 10.3390/medicines7090048

**Published:** 2020-08-25

**Authors:** Chiaki Uehara, Nobuyuki Miyatake, Shuhei Hishii, Hiromi Suzuki, Akihiko Katayama

**Affiliations:** 1Department of Hygiene, Faculty of Medicine, Kagawa University, Miki, Kagawa 761-0793, Japan; miyarin@med.kagawa-u.ac.jp (N.M.); duratech117@yahoo.co.jp (S.H.); tanzuki@med.kagawa-u.ac.jp (H.S.); 2Department of Nursing, Kagawa Prefectural University of Health Sciences, Takamatsu 761-0123, Japan; 3The Faculty of Social Studies, Shikokugakuin University, Zentsuji, Kagawa 765-8505, Japan; kata@sg-u.ac.jp

**Keywords:** continuous sedentary behavior (CSB), sedentary behavior (SB), tri-accelerometer, body mass index (BMI), community-dwelling Japanese adults

## Abstract

**Background:** Sedentary behavior (SB) is associated with adverse health outcomes. The aim of this study was to clarify seasonal changes in SB including continuous SB (CSB) in community-dwelling Japanese adults. **Methods:** In this secondary analysis, a total of 65 community-dwelling Japanese adults (7 men and 58 women, 69 (50–78) years) were enrolled. SB (%), including CSB (≥30 min) as well as physical activity, were evaluated using a tri-accelerometer. The differences in these parameters between baseline (summer) and follow-up (winter) were examined. **Results:** %CSB and %SB at baseline were 20.5 (4.0–60.9) and 54.0 ± 11.5, respectively. CSB was significantly increased (6.6%), and SB was also increased (5.1%) at follow-up compared with baseline. In addition, there were positive relationships between changes in CSB and SB, and body weight and body mass index. **Conclusions:** These results suggest that there were significant seasonal changes in CSB and SB in community-dwelling Japanese adults.

## 1. Introduction

Sedentary behavior (SB), defined as ≤1.5 metabolic equivalents (METs) during waking hours [1], is closely associated with health outcomes such as cardiovascular diseases [2] and death [3,4]. The results of linear regression analysis between SB and cardiometabolic risk variables showed that time spent on SB was significantly associated with decreased HDL cholesterol, one of the risks of cardiovascular disease [β = −0.02 (95%CI: −0.02; −0.01)] [2]. In a prospective cohort study investigating the association between SB and mortality risk, subjects with a longer SB (quartile 4) showed a higher all-cause mortality rate than those with a shorter SB (quartile 1) (Hazard ratio = 2.63 (95% CI: 1.60–4.30)) [3]. Biswas et al. reported, by using systematic reviews and meta-analysis, that prolonged SB adversely affects health outcomes such as diabetes and cancer incidence and all-cause mortality, independent of physical activity [4]. We have also proved that SB is closely linked to health-related quality of life (HRQOL) and all-cause mortality in patients on chronic hemodialysis [5,6]. Recently, the prevalence of SB in adults was reported to be about 60% [7] and it has increased because of modern lifestyles [8]. The rate of SB in Japan is reported to be the highest among 20 countries by the descriptive epidemiology using the International Physical Activity Questionnaire (IPAQ). Specifically, the SB time (median) in 20 countries was 300 min/day, while in Japan was 420 min/day [9].

In addition, not only total SB but also continuous SB (CSB) (Defined as ≥30 min) are focused on and is closely related to health outcomes [10,11,12]. Harley et al. pointed out the importance of assessing CSB, which does not include a break, as well as total SB [10]. Peddie et al. also found that taking a break every 30 min during CSB was effective for lowering postprandial blood glucose [11]. In a prospective cohort study of Japanese adults, longer time spent in prolonged sedentary bouts (≥30 min) was significantly associated with a higher risk of metabolic syndrome [12]. These results suggest that evaluating CSB in addition to SB is important in clinical practice.

Many factors have a clinical impact on SB including disease [4], psychological distress [13], sleeping time [14], anthropometric parameters [15,16], sex, age [16,17], education, occupation, marriage [18,19], and social participation [18,20]. Although there were some reports of a relationship between SB and environmental factors including air temperature [21] and daylight hours [22], the effect of seasonal changes in SB [23,24,25], especially CSB, has not been fully investigated.

Seasonal changes are associated with health outcomes [26,27]. Bnado et al. reported that lower air temperature associated with increased mortality including heart disease and renal failure in Japan [26]. Sakamoto et al. reported that diabetes worsens in winter [27] and that the levels and total amount of physical activity are affected by meteorological factors [28]. Tucer et al. suggested by a systematic review that seasonal changes impair physical activity levels and population participation in physical activity, and they proposed to examine seasonal effects to promote physical activity [29]. Therefore, it is important to consider seasonality for promoting health. However, it is difficult to encourage elderly subjects and patients who are physically weak to maintain and improve their physical activity. Therefore, we focused on the seasonality of SB and CSB because reducing SB including CSB is thought to be easier than increasing physical activity. There are no studies that have evaluated the seasonality of SB and CSB in Japanese adults.

Taken together, in this secondary analysis, we investigated the seasonal changes in CSB and SB in community-dwelling Japanese adults.

## 2. Subjects and Methods

### 2.1. Subjects

This study consisted of a secondary analysis following the first analysis from May 2016 to December 2016 as previously described [30]. “Participants voluntarily took part in the previously reported randomized controlled trial (RCT) of an exercise program who lived around Seto Inland Sea, Western Japan [31]. The primary outcome was the general health questionnaire-12 (GHQ-12) between the two groups with the difference (0.7) and standard deviation (SD) (1.2) (α = 0.05, β = 0.20, dropout rate = 10%). The sample size of the first study was 120.” A total of 65 community-dwelling Japanese adults (7 men and 58 women, 69 (50–78) years) were selected for the secondary analysis according to the following criteria (Figure 1): (1) they underwent the measurements of CSB and SB including clinical parameters on August 2017, (2) they provided written informed consent.

Ethical approval was obtained from the ethics committee of Shikokugakuin University (Approved number: 2015001, Date: 26 May 2015).

### 2.2. Clinical Parameters

Clinical parameters of enrolled subjects were sex, age, height (cm), body weight (kg), and body fat percentage (%). The height and body weight were measured by well-trained medical staff. Body mass index (BMI) was calculated as follows: body weight (kg)/(height (m))^2^. Body fat percentage was measured using the impedance method (MC-180, Tanita Co. Ltd., Tokyo, Japan) as previously described [32]. Lean body mass (kg) was calculated as follows: body weight (kg) − [total body weight (kg) × body fat percentage (%)/100]. These data were measured at RCT baseline.

### 2.3. Tri-Accelerometer Measurements

CSB, SB, and physical activity parameters, such as daily step counts (steps/day), light-intensity physical activity (LPA) (1.6–2.9 METs), and moderate-to-vigorous physical activity (MVPA) (≥3.0 METs), were measured using a tri-accelerometer (Active Style Pro HJA-350IT, Omron Health Company, Kyoto, Japan) for seven days [5]. Active Style Pro has been verified for reproducibility and reliability as previously described [33,34]. Furthermore, many of the SB studies in Japan use this tri-accelerometer [35,36]. Subjects were required to wear on their waist its tool from morning to night, except when sleeping; it was not possible to use while participating in underwater activities (bathing and swimming) and during sports activities that may involve contact. The standard deviation (SD) of the data of 60 s was defined as an average value of acceleration. The average values of three days when the tri-accelerometer was worn for 600 min or more per day were used for analysis [37,38]. %CSB was defined as CSB = (time of sedentary behavior continuous for more than 30 min per day)/(awake time in one day; minutes) × 100.

### 2.4. Metrological Parameters

Mean air temperature, total precipitation, mean humidity, and daylight hours per month in summer (August) were 29.0 °C, 89.5 mm, 71.0%, and 251.1 h. Mean air temperature, total precipitation, mean humidity, and daylight hours per month in winter (February) were 5.2 °C, 37.5 mm, 64.0%, and 170.2 h, respectively. Furthermore, the highest temperature in summer (August) was 38.2 °C and the lowest in winter (February) was −3.6 °C. During tri-accelerometer measurements, several hot days (≥35 °C) and winter days (<0 °C) were recorded [39].

### 2.5. Statistical Analysis

Data were distributed both normally and non-normally, which were evaluated by using the Shapiro–Wilk test. Data were expressed as the mean ± standard deviation (SD) (normal distribution) and median (minimum–maximum) (non-normal distribution). Comparison parameters between subjects with and without follow-up at baseline were analyzed by unpaired *t* test, Mann–Whitney U test, and covariance analysis. The difference of parameters between baseline (summer) and follow-up (winter) were evaluated by paired *t* test and Wilcoxon signed-rank test. Simple and partial correlation and Spearman’s rank correlation coefficient analyses between changes in CSB and SB, and clinical parameters at baseline (summer), were employed. *p* < 0.05 was significant. All statistical analyses were performed using JMP 13.2 software (SAS, Cary, NC, USA).

## 3. Results

Table 1 shows the clinical characteristics of enrolled subjects in this secondary analysis at baseline (summer). %CSB and %SB were 20.5 (4.0–60.9) and 54.0 ± 11.5, respectively. Comparison of clinical parameters between subjects with and without follow-up at baseline are summarized in Table 2. Among 24 subjects without follow-up, CSB and SB could not be evaluated in 16 subjects because of the difficulty of wearing a tri-accelerometer. Clinical parameters including age, anthropometric, and body composition parameters were not different between the two groups. In addition, there were no significant differences in CSB, SB, or physical activity parameters between subjects with and without follow-up at baseline. No differences in CSB and SB were observed even after adjusting for sex, age, and BMI by covariance analysis.

We evaluated the changes in parameters between baseline and follow-up (Table 3). There were significant positive changes in CSB (6.6%) and SB (5.1%) at follow-up. LPA and MVPA were significantly decreased. However, there was no significant difference in the daily step counts.

Finally, we evaluated the relationships between changes in CSB, SB, and clinical parameters at baseline (Table 4). There were significant positive relationships between changes in CSB and SB, and body weight and BMI. The relationships between changes in CSB and SB, and BMI remained even after adjusting for sex. However, there were no relationships between changes in CSB or SB, and clinical parameters i.e., age, height, body fat percentage, and LBM.

## 4. Discussion

In this study, we evaluated the differences in CSB and SB between baseline (summer) and follow-up (winter) in community-dwelling Japanese adults. We found that CSB and SB were significantly increased in the winter compared with the summer.

There have been some reports on weather conditions and SB. For example, in a prospective cohort study using a tri-accelerometer, Sartini et al. showed that older British men were sedentary 26 min more per day at a lower air temperature compared with a higher air temperature [21]. Schepps et al. reported that older women in the United States with ≥14 h daylight hours spent 1.6% less time in SB compared with those with <10 h daylight hours [22]. Both elderly community-dwelling men and women in Iceland spent less time in SB (4.4% in men, 2.5% in women) in summer compared with winter [23]. In Japan, Tanaka et al. reported that daily SB was significantly higher during the summer vacation (July and August) than during the school term (May) for both boys and girls [25]. Since the climate of Japan is in the mid-latitude, the seasonal difference is large [39,40]. In Japan, air temperature is lower in winter than in summer, and the daylight hours are shorter. Therefore, in this study as well as in the previous study, sedentary behavior decreased significantly from summer to winter.

In our study, we found that SB, especially CSB, at follow-up was significantly longer than that at baseline in community-dwelling Japanese adults. An increase of 6.6% in CSB at follow-up compared with baseline was noted. This corresponded to at least 39.6 min per day. Although there is no report of a relationship between increased CSB/day and health outcomes, Fishman et al. found that replacing 30 min of sedentary time by LPA or MVPA was associated with a significant reduction in mortality risk in a prospective cohort study [41]. Parsons et al. also reported that each additional 30 min of SB per day was associated with a significant higher odds ratio of lower estimated glomerular filtration rate (eGFR) in a cross-sectional population-based study [42]. In addition, there was a significant relationship between changes in CSB and SB, and body weight and BMI at baseline, suggesting that obese people might become more inactive in winter. There have been some reports that obesity is closely associated with SB. In a study using a linear regression model, midlife obesity was significantly associated with sedentary time per day in old age [15]. Chen et al. reported that those with a high BMI had a low moderate to intense physical activity [16]. Obese women showed longer SB in the evening (6 pm–midnight) rather than in the morning (6 am–noon) and afternoon (noon–6 pm) [43]. Taken together, we recommend reducing SB, and especially CSB in winter, especially in subjects with a higher body weight and/or BMI. In addition, we also have to pay attention to the seasonal differences in CSB and SB when we evaluate CSB and SB in clinical practice. Considering seasonality of CSB and SB will provide us with more viable and effective information for health education.

On the other hand, daily step counts (steps/day) did not significantly decrease from summer to winter. Hamilton et al. reported that daily step counts of adults in the UK were significantly reduced in winter compared to summer on all days [44]. However, in a large-scale cohort study in Japan for people over 40 years old, daily step counts peaked at temperatures 19.4 to 20.7 °C and at temperatures higher than those at which step counts peaked, the decrease in steps per day was 98.0 to 187.9 per 1 °C increase. Especially, the step count of elderly people was more affected by a decrease in steps per day when the value of the metrological parameters was higher than that of the lower value [45]. This study inferred that one of the reasons that daily step counts did not decrease significantly from summer to winter was because the number of steps in summer was affected by the heat wave in Japan as well. Even considering daily step counts (steps/day), we recommend reducing SB, and especially CSB, in winter.

There were some limitations in this study. First, this was a secondary small sample size study. The number of male participants was especially low (n = 7). Japanese men have little social interaction with others and have low sociality compared to women. Therefore, they tend not to participate in community health projects, especially in daytime. Similarly, in this study, there were few male participants [46]. In addition, there may be difficulty in wearing a tri-accelerometer for at least three days at every measurement. Sixteen subjects could not wear the tri-accelerometer in winter. Secondly, although this secondary analysis was conducted eight months after the first study finished, there may be a legacy effect of the first study on these results, especially on CSB and SB. Third, enrolled subjects in this study were thought to be more health conscious than average people. Fourth, factors contributing to the difference in CSB and SB between baseline and follow-up were not investigated. Fifth, we did not accurately evaluate the type of activity during tri-accelerometer measurements. Nevertheless, we found that CSB and SB at follow-up were increased compared with those at baseline in community-dwelling Japanese adults. Further studies with a larger sample size are required.

## Figures and Tables

**Figure 1 medicines-07-00048-f001:**
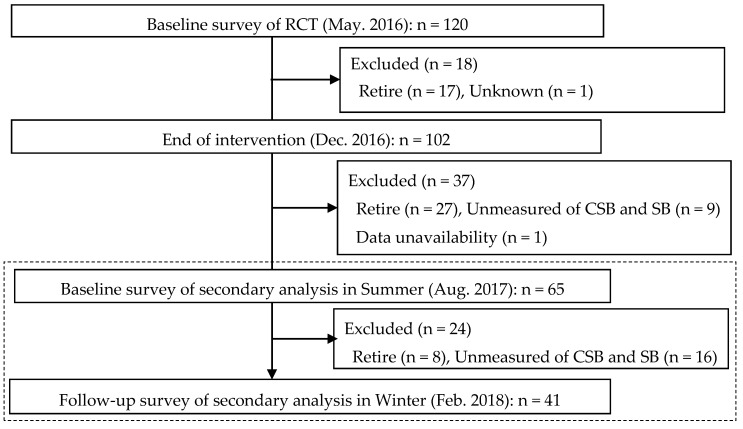
Flow diagram of analysis.

**Table 1 medicines-07-00048-t001:** Clinical characteristics of enrolled subjects at baseline (summer).

	Total (n = 65)		Men (n = 7)		Women (n = 58)	
Age (years)	69	(50–78) ^b^	72.1	±5.3 ^a^	69	(50–78) ^b^
Height (cm)	154.0	(138.0–175.5) ^b^	166.6	±6.5 ^a^	153.5	±5.1 ^a^
Body weight (dry weight) (kg)	52.4	(41.2–74.3) ^b^	58.4	±5.4 ^a^	51.9	(41.2–74.3) ^b^
BMI (kg/m^2^)	22.0	±2.1 ^a^	20.6	(20.1–21.9) ^b^	22.2	±2.2 ^a^
Body fat percentage (%)	26.8	±7.3 ^a^	13.1	±3.2 ^a^	28.4	±5.8 ^a^
LBM (kg)	37.4	(29.0–60.4) ^b^	50.7	±5.0 ^a^	37.1	±2.6 ^a^
Daily step counts (steps/day)	6350.6	±2292.7 ^a^	8207.3	±1210.6 ^a^	6126.6	±2296.3 ^a^
SB (%)	54.0	±11.5 ^a^	55.6	±13.1 ^a^	53.8	±11.4 ^a^
CSB (%)	20.5	(4.0–60.9) ^b^	25.0	±13.4 ^a^	20.3	(4.0–60.9) ^b^
LPA (%)	40.1	±10.5 ^a^	34.9	±9.5 ^a^	40.8	±10.5 ^a^
MVPA (%)	6.1	(1.3–25.5) ^b^	8.0	(5.9–25.5) ^b^	5.6	(1.3–17.8) ^b^

^a^: Values evaluated as normal distribution using Shapiro–Wilk test are shown as the mean ± standard deviation. ^b^: Values evaluated as non-normal distribution using a Shapiro–Wilk test are shown as the median (minimum–maximum). BMI: body mass index, LBM: lean body mass, SB: sedentary behavior, CSB: continuous SB, LPA: light-intensity physical activity (1.6–2.9 METs), MVPA: moderate-to-vigorous physical activity (≥3.0 METs).

**Table 2 medicines-07-00048-t002:** Comparison of clinical parameters between subjects with and without follow-up at baseline (summer).

	Follow-Up (+) (n = 41)		Follow-Up (−) (n = 24)		*p*	*p* ^1^
**Men/Women**	**5/36**		**2/22**			
Age (years)	68.7	±5.6 ^a^	67.9	±6.6 ^a^	0.577 ^c^	
Height (cm)	154.0	(148.0–173.0) ^b^	153.6	±7.3 ^a^	0.475 ^d^	
Body weight (dry weight) (kg)	52.9	±5.0 ^a^	52.1	(41.2–74.3) ^b^	0.519 ^d^	
BMI (kg/m^2^)	21.8	±1.9 ^a^	22.1	(18.6–28.6) ^b^	0.624 ^d^	
Body fat percentage (%)	27.1	(9.5–35.7) ^b^	28.1	±8.1 ^a^	0.373 ^d^	
LBM (kg)	38.0	(32.5–53.6) ^b^	36.6	(29.0–60.4) ^b^	0.093 ^d^	
Daily step counts (steps/day)	6491.8	±1892.8 ^a^	6109.4	±2880.8 ^a^	0.521 ^c^	
SB (%)	55.2	±11.2 ^a^	52.0	±11.9 ^a^	0.285 ^c^	0.397
CSB (%)	24.7	±12.2 ^a^	19.5	(5.2–60.9) ^b^	0.209 ^d^	0.320
LPA (%)	39.3	±10.3 ^a^	41.5	±10.8 ^a^	0.408 ^c^	
MVPA (%)	6.1	(1.3–15.3) ^b^	6.2	(1.8–25.5) ^b^	0.573 ^d^	

^a^: Values evaluated as normal distribution using Shapiro–Wilk test are shown as the mean ± standard deviation. ^b^: Values evaluated as non-normal distribution using Shapiro–Wilk test are shown as the median (minimum–maximum). BMI: body mass index, LBM: lean body mass, SB: sedentary behavior, CSB: continuous SB, LPA: light-intensity physical activity (1.6–2.9 METs), MVPA: moderate-to-vigorous physical activity (≥3.0 METs). *p*: Significance was determined using analysis of unpaired *t* test (c) and Man–Whitney U test (d). *p*^1^: Significance was determined using analysis of covariance (ANCOVA) adjusted for sex, age, and BMI.

**Table 3 medicines-07-00048-t003:** Changes in physical activity and sedentary behavior between baseline (summer) and follow-up (winter).

	Baseline (Summer)		Follow-Up (Winter)		Difference	95%CI	*p*
					(Winter–Summer)	(Lower, Upper)	
Daily step counts (steps/day)	6491.8	±1892.8 ^a^	5751.5	±2768.4 ^a^	−740.4	(−1530.1, 49.4)	0.065 ^c^
SB (%)	55.2	±11.2 ^a^	60.2	±10.4 ^a^	5.1	(1.9, 8.3)	**0.003** ^c^
CSB (%)	24.7	±12.2 ^a^	31.4	±14.7 ^a^	6.6	(2.4, 10.8)	**0.003** ^c^
LPA (%)	39.3	±10.3 ^a^	35.2	±9.7 ^a^	−4.1	(−6.9, −1.3)	**0.006** ^c^
MVPA (%)	6.1	(1.3–15.3) ^b^	4.9	(0.8–13.9) ^b^	−1.2	(−2.0, −0.4)	**0.002** ^d^

^a^: Values evaluated as normal distribution using Shapiro–Wilk test are shown as the mean ± standard deviation. ^b^: Values evaluated as non-normal distribution using Shapiro–Wilk test are shown as the median (minimum–maximum). ^c^: Significance was determined using analysis of paired *t* test. ^d^: Significance was determined using analysis of Wilcoxon signed-rank test. Bold values indicate statistically significant (*p* < 0.05). BMI: body mass index, LBM: lean body mass, SB: sedentary behavior, CSB: continuous SB, LPA: light-intensity physical activity (1.6–2.9 METs), MVPA: moderate-to-vigorous physical activity (≥ 3.0 METs), CI: confidence interval.

**Table 4 medicines-07-00048-t004:** Relationship between changes in SB and CSB, and clinical parameters at baseline (summer).

	ΔSB				ΔCSB			
	*r*	*p*	*r* ^1^	*p* ^1^	*r*	*p*	*r* ^1^	*p* ^1^
Age (years) ^a^	−0.034	0.832			0.166	0.299		
Height (cm) ^a^	−0.021	0.894			−0.013	0.938		
Body weight (dry weight) (kg) ^a^	0.398	**0.010**			0.434	**0.005**		
BMI (kg/m^2^) ^b^	0.381	**0.014**	0.432	**0.005**	0.356	**0.022**	0.418	**0.009**
Body fat percentage (%) ^b^	0.278	0.078			0.261	0.099		
LBM (kg) ^a^	0.090	0.574			0.075	0.641		

Δ: changes in parameters between baseline (summer) and follow-up (winter). BMI: body mass index, LBM: lean body mass. ^a^: Spearman’s rank correlation coefficient. ^b^: Pearson’s correlation coefficient. *r*
^1^: partial correlation coefficient adjusting for sex.

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
