# Peer review of "Seasonal Changes in Continuous Sedentary Behavior in Community—Dwelling Japanese Adults: A Pilot Study"

_medicines, 2020, doi:10.3390/medicines7090048_

Round 1
Reviewer 1 Report
An interesting study, but with a very small sample; generalizability is limited. The findings are not very surprising; people move less (sit still longer) when it’s colder. Still, I think sedentary behavior researchers might be interested.
I have four minor points that need clarification:
Could authors explain (or speculate) a little more on the reasons for the seasonal differences seen here? Air temperature and daylight hours are mentioned (referring to other studies), but how big is the difference in temperature and daylight in Japan, between summer and winter? Also, could the reduction simply be explained by return to work? (Assuming that most Japanese take vacation in summer and work in winter?).
This is a secondary analyses; how did the exercise intervention affect these results? If sedentary adults were recruited in winter, then encouraged to exercise (for how long?) and assessed in winter, I might expect some effect of the exercise on sedentary time (i.e. that it might reduce). Could authors briefly comment?
I am not familiar with the specific tri-axial accelerometer used, but probably, it did not measure sedentary behavior as well as, for example, the ActiPal, which is an inclinometer also. Could authors comment on the accuracy of this device in measuring sedentary behavior specifically (as opposed to LPA/MVPA). Accelerometers are considered gold standard, but still miss some activities (swimming, weight training, even cycling).
Related to the last point, accelerometers can’t tell us what type of activity was done, and this is relevant because recent studies suggest differential associations of passive (e.g. TV viewing) and mentally-active (e.g. reading) SB with health outcomes. I suggest this is a study limitation.
Thank you authors.
Author Response
August 19, 2020
Dear Reviewer 1
We are grateful for the time and energy you expended to review our paper manuscript ID: medicines-866553 entitled “Seasonal Changes in Continuous Sedentary Behavior in Community-Dwelling Japanese Adults: A Pilot Study”. Your comments have been helpful to allowing us to revise and strengthen our manuscript. Especially, we could strengthen the explanation of a tri-accelerometer. In addition, we could strengthen the explanation of Japan's climate and the seasonality of continuous sedentary behavior (CSB).
The detail review of the manuscript is appreciated and we have attempted to answer each of the questions raised. The revised manuscript is submitted.
Please see the attachment.
Sincerely,
Corresponding Author name: Chiaki Uehara

Reviewer 2 Report
I enjoyed reading the article. The analysis seemed valid given the data presented. The limitations of the research were communicated well in both the introduction and the conclusion.
I found it interesting that steps/ day did not significantly decrease from Summer to Winter. The data is in the right direction, but it would have been nice to include a sentence or 2 in the conclusion theorizing why there wasn't as big an effect with steps / day compared to all the other measures. Is this there a possible cultural explanation or could similar results regarding steps be expected in other older cohorts?
Another minor detail, I couldn't find any information on how the accelerometers were worn. This may have been included in the description of the original study but it would be useful to include in this paper for clarity.
Author Response
August 19, 2020
Dear Reviewer 2
We are grateful for the time and energy you expended to review our paper manuscript ID: medicines-866553 entitled “Seasonal Changes in Continuous Sedentary Behavior in Community-Dwelling Japanese Adults: A Pilot Study”. Your comments have been helpful to allowing us to revise and strengthen our manuscript. Especially, we could strengthen the explanation of steps/ day why did not significantly decrease from Summer to Winter. In addition, we could strengthen the explanation of a tri-accelerometer.
The detail review of the manuscript is appreciated and we have attempted to answer each of the questions raised. The revised manuscript is submitted.
Please see the attachment.
Sincerely,
Corresponding Author name: Chiaki Uehara
